# Influence of Malignant Pleural Fluid from Lung Adenocarcinoma Patients on Neutrophil Response

**DOI:** 10.3390/cancers14102529

**Published:** 2022-05-20

**Authors:** Maria Mulet, Rubén Osuna-Gómez, Carlos Zamora, José M. Porcel, Juan C. Nieto, Lídia Perea, Virginia Pajares, Ana M. Muñoz-Fernandez, Nuria Calvo, Maria Alba Sorolla, Silvia Vidal

**Affiliations:** 1Inflammatory Diseases, Institut de Recerca de l’Hospital de la Santa Creu i Sant Pau, Biomedical Research Institute Sant Pau (IIB Sant Pau), 08041 Barcelona, Spain; mmulet@santpau.cat (M.M.); rosuna@santpau.cat (R.O.-G.); carlosza86@gmail.com (C.Z.); juan.nieto@cnag.crg.eu (J.C.N.); perasoriano.lidia@gmail.com (L.P.); 2Pleural Medicine Unit, Department of Internal Medicine, Hospital Universitari Arnau de Vilanova, IRBLleida, University of Lleida, 25003 Lleida, Spain; jmporcel.lleida.ics@gencat.cat; 3Department of Pneumology, Hospital de la Santa Creu i Sant Pau, 08041 Barcelona, Spain; vpajares@santpau.cat (V.P.); amunozf.girona.ics@gencat.cat (A.M.M.-F.); 4Department of Oncology, Hospital de la Santa Creu i Sant Pau, 08041 Barcelona, Spain; ncalvo@santpau.cat; 5Research Group of Cancer Biomarkers, IRBLleida, 25198 Lleida, Spain; msorolla@irblleida.cat

**Keywords:** malignant pleural fluid, lung adenocarcinoma, neutrophils, NETosis

## Abstract

**Simple Summary:**

This study provides novel information about the role of neutrophils in malignant pleural effusion (MPE) and hallmarks their clinical relevance. Since these cells have emerged as important regulators of cancer, we characterized their phenotype and functions in MPE microenvironment. We found that neutrophil-derived products (degranulation molecules and neutrophil extracellular traps (NETs)) were increased in MPE. In addition, NETs were associated with a worse outcome in lung adenocarcinoma patients with MPE.

**Abstract:**

Malignant pleural effusion (MPE) is a common severe complication of advanced lung adenocarcinoma (LAC). Neutrophils, an essential component of tumor infiltrates, contribute to tumor progression and their counts in MPE have been associated with worse outcome in LAC. This study aimed to evaluate phenotypical and functional changes of neutrophils induced by MPE to determine the influence of MPE immunomodulatory factors in neutrophil response and to find a possible association between neutrophil functions and clinical outcomes. Pleural fluid samples were collected from 47 LAC and 25 heart failure (HF) patients. We measured neutrophil degranulation products by ELISA, oxidative burst capacity and apoptosis by flow cytometry, and NETosis by fluorescence. The concentration of degranulation products was higher in MPE-LAC than in PE-HF. Functionally, neutrophils cultured with MPE-LAC had enhanced survival and neutrophil extracellular trap (NET) formation but had reduced oxidative burst capacity. In MPE, NETosis was positively associated with MMP-9, P-selectin, and sPD-L1 and clinically related to a worse outcome. This is the first study associating NETs with a worse outcome in MPE. Neutrophils likely contribute to tumor progression through the release of NETs, suggesting that they are a potential therapeutic target in LAC.

## 1. Introduction

Malignant pleural effusion (MPE), a common complication of advanced cancer, is defined by the presence of tumor cells in pleural fluid and is associated with poor prognosis [1]. Metastatic lung cancer has been identified as the cause of over one-third of all MPEs. Non-malignant conditions, such as congestive heart failure (HF), can also lead to the build-up of fluid in the pleural space due to increased hydrostatic pressure [2]. However, some limitations hamper both an accurate diagnosis and the discrimination between MPEs and benign pleural effusions (BPEs) [3]. Considering that multiple factors contribute to the complexity in MPE diagnosis, the identification of definitive biomarker requires to unravel the molecular and cellular composition of these pleural fluids.

The cellular composition of pleural effusions depends on the etiology of the disease and may be related with the outcome. In particular, a high neutrophil-to-lymphocyte ratio (NLR) in MPE has been associated with shorter overall survival in lung cancer patients [4]. In line with these findings, we have recently found a negative association between the presence of neutrophils in the pleural fluid and the survival of lung adenocarcinoma (LAC) patients with MPE [5]. Although our previous study showed no differences in the number of neutrophils between MPEs and BPEs, we also reported an increased neutrophil count in those MPEs with more tumor cells, suggesting a possible link between those two cell types [5]. It is possible that circulating granulocytes rapidly migrate to the site of inflammation (i.e., the pleural space) under the influence of several chemoattractants such as interleukin-8 (IL-8) [6,7], leukotriene B4 [8], and epithelial neutrophil-activating peptide-78 [9].

In addition to their traditionally antimicrobial function, the role of neutrophils in the tumor microenvironment (TME) as immunosuppressor cells has been increasingly appreciated [10]. Recently, Singel et al. observed a neutrophil suppressor phenotype when co-culturing them with stimulated T cells in the presence of MPEs [11]. The polarization of neutrophils toward a protumor or antitumor phenotype depends on inflammatory mediators secreted by the tumor and immune cells of the TME [12]. As protumor cells, neutrophils promote LAC growth via the secretion of neutrophil elastase (NE) [13] and metastasis through the release of matrix metalloproteinase-9 (MMP-9), which degrades the basement membrane [14]. Another mechanism related to tumor progression is the formation of neutrophil extracellular traps (NETs). Diverse stimuli are thought to be involved in the induction of NETs, including microorganisms, activated platelets [15], and proinflammatory cytokines [16]. NETs lead to a cell death known as NETosis. During NETosis, there is a release of decondensed deoxyribonucleic acid (DNA), accompanied by citrullinated histones and granule proteins [17]. Neutrophils can produce NETs, which would protect tumor cells from cytotoxic T lymphocytes [18] and awaken dormant cancer cells [19]. As antitumor cells, neutrophils mediate tumor cell death though the production of reactive oxygen species (ROS) [20] and nitric oxide (NO) [21].

Based on our previous results, we have considered two possible nonexclusive mechanisms to explain the role of neutrophils in MPE. First, we propose that phenotypic changes on neutrophils, probably due to the presence of a tumor, regulate cell function, and promote tumor progression. Second, soluble molecules in MPE can modulate neutrophil response against cancer, including viability, degranulation, NETosis, and oxidation burst. Considering the importance of platelet–neutrophil crosstalk in cancer, we also analyze the association between platelet-derived factors and neutrophil responses. Finally, we determined the clinical relevance of neutrophil activity as a diagnostic or/and prognostic biomarker for LAC patients with MPE.

## 2. Materials and Methods

### 2.1. Patients and Pleural Fluid Samples

We prospectively enrolled patients with MPE secondary to LAC (stage III/IV) (*n* = 47) and patients with BPE due to congestive HF (*n* = 25) who were admitted to Hospital de la Santa Creu i Sant Pau (Barcelona) and Hospital Arnau de Vilanova (Lleida) (Table 1). Malignancy of pleural fluids secondary to LAC was based on the cytological detection of tumor cells. Congestive HF is a chronic progressive condition that alters the pumping capacity of the heart muscle. PEs from HF patients were used as a negative control of malignancy. Pleural fluid samples were collected at the time of MPE diagnosis, in the case of LAC patients, by thoracentesis and 10 U/mL heparin (Hospira, Lake Forest, IL, USA) was added. Pleural fluid samples were filtered before spinning them down to cell pellets. Cell composition was subsequently analyzed by flow cytometry and cell-free pleural effusions (cf-PEs) were filtered again (0.22 µm) and stored at −80 °C until use. The study was approved by the Institutional Ethics Committee of the Hospital de la Santa Creu i Sant Pau.

### 2.2. Phenotyping Neutrophils from Pleural Effusions by Flow Cytometry

Cells from pleural fluids were collected by centrifugation and red blood cells were lysed. Then, one million cells were stained with different mAb panels to identify the neutrophil population and to determine the surface expression of inhibitory receptors in a MACSQuant cytometer. Antibody panels included anti-CD45-VioBlue (Miltenyi Biotec, Bergisch Gladbach, Germany, clone 8C11), anti-CD14-APC (Immunotools, Friesoythe, Germany, clone OKT-4), anti-CD16-FITC (Immunotools, clone 3G8) and inhibitory receptors markers. In Panel A, cells were stained with anti-CD200R-PerCP (eBiosecience, San Diego, CA, USA, clone OX108). In Panel B, cells were stained with anti-CD85a-PerCP-Vio770 (Miltenyi Biotec, clone REA207). Panel C included anti-CD14-APC, -CD15-PE-Cy7 (BD Bioscience, San Jose, CA, USA, clone HI98), -S100A8/9-FITC (OriGene Technologies Inc, Rockville, MD, clone BM4025F) and -HLA-DR-APC-H7 (BD Biosciences, clone G46-6). Finally, cells were washed with PBS 1% bovine serum albumin (BSA), resuspended in PBS, and acquired by flow cytometry. Neutrophils were identified as CD14^-^ CD16^+^ or CD14^-^ CD15^+^ cells after being gated on CD45/side-scatter (SSC) plot. The percentage of positive cells and the integrated mean fluorescence intensity (iMFI) of inhibitory receptors were calculated using FlowJo software version 10.6.1 (Ashland, OR, USA). The identification and quantification of tumor cells in MPEs from LAC patients was assessed by flow cytometry [5].

### 2.3. Isolation of Neutrophils from Healthy Donors

Venous blood from 10 healthy donors was taken from the blood bank of Hospital de la Santa Creu i Sant Pau and collected in heparin-coated Vacutainer tubes (BD Bioscience). Neutrophils were isolated by Ficoll-Hypaque (Lymphoprep Axis-shield PoCAs, Oslo, Norway) density-gradient centrifugation followed by dextran-sucrose (Sigma-Aldrich, St. Louis, Missouri, USA) sedimentation for 30 min. Finally, red blood cells were lysed (RBC Lysis Buffer, BioLegend, San Diego, CA, USA) and purity >95% was assessed by flow cytometry.

### 2.4. Induction and Quantification of NETosis in the Presence of Cell-Free Pleural Fluids

Briefly, freshly isolated healthy neutrophils were resuspended at 6 × 10^6^ cell/mL in complete RPMI medium supplemented with 25% of cf-MPE-LAC or cf-PE-HF. We also cultured cells with completed RPMI medium as a positive control for NETosis. Then, neutrophils were seeded into a black 96-well plate and stimulated with 25nM phorbol-12-mystrate-13-acetate (PMA; Sigma-Aldrich) for three hours in a humidified incubator (37 °C) or remained unstimulated as a control. Next, cells were collected by centrifugation and supernatant was stained with Sytox^TM^ Green nucleic acid stain (5 µM; Invitrogen, Eugene, OR, USA) for 20 min. The relative fluorescence was then read with a fluorometer Infinite M200 Pro Microplate reader (Tecan Group, Männedorf, Switzerland) with a filter setting of 485 nm (excitation)/520 nm (emission). Fluorescence intensity reflects the amount of DNA. Results were expressed as relative fluorescence units (RFUs). Endogenous pleural DNA was measured by basal pleural emission (without cells) and subtracted from pleural fluid culture wells.

### 2.5. Measurement of Oxidative Burst in the Presence of Cell-Free Pleural Fluids

Isolated neutrophils from healthy donors (2 × 10^4^ cells) were cultured in the presence of HBSS medium containing 25% of cf-MPE-LAC, 25% cf-PE-HF or 2% BSA as a control [22]. Briefly, cells were first incubated in the presence of dihydrorhodamine 123 (DHR; Sigma-Aldrich) for 10 min and then stimulated with PMA (325 nM; Sigma Aldrich) for 30 min or remained unstimulated in a humidified incubator (37 °C). ROS production was measured by oxidation of DHR to rhodamine by flow cytometry. Data were expressed as oxidative burst indexes (MFI of DHR under pleural fluid culture conditions divided by the respective nonpleural fluid culture control).

### 2.6. Induction of Neutrophil Apoptosis in the Presence of Cell-Free Pleural Fluids

Healthy neutrophils (1 × 10^6^ cells) were cultured in RPMI medium supplemented with 25% of cf-MPE-LAC or cf-PE-HF for 24 h. Neutrophils (1 × 10^6^ cells) were also cultured in RPMI to have spontaneous apoptosis. After 24 h, neutrophils were stained with anti-CD66b-PE mAb (BioLegend, clone G10F5) for 15 min and washed with PBS. Subsequently, cells were resuspended in 100 µL of Annexin V Binding Buffer (10 mM HEPES, pH 7.4, containing 140 mM NaCl and 2.5 mM CaCl_2_) and stained with 5 µL of Annexin V-FITC (Immunotools) and 5 µL propidium iodide (PI, Invitrogen, Eugene, OR) 15 min at RT in the dark. Cells were finally washed with PBS and resuspended in 100 µL for cytometry acquisition. Viable neutrophils were double negative for Annexin-V and PI. Results were analyzed using FlowJo software version 10.6.1.

### 2.7. Determination of Biochemical Parameters and Soluble Mediators in Pleural Fluids

Protein and glucose levels, pH, lactate dehydrogenase (LDH), C-reactive protein (CRP), and adenosine deaminase (ADA) measurements in pleural fluids were obtained from routine clinical analysis. Pleural fluid concentration of soluble P-selectin (R&D Systems MN, USA), PD-L1 (Thermo Fisher Scientific, Vienna, Austria), TGF-β1 (Mabtech, Nacka Strand Sweden), PF4 (Peprotech, Rocky Hill, NJ, USA), VEGF (Peprotech), CD40L (Peprotech), IL-8 (Mabtech, Nacka Strand, Sweden), MPO (Cloud-Clone, Wuhan, China), lactoferrin (RayBiotech, Peachtree Corners, GA, USA) and MMP-9 (Boster Bio, Pleasanton, CA, USA) were determined by specific ELISAs using standard curves according to the manufacturer’s instructions of ELISA kit. Pleural fluids were measured for nitrate and nitrite species using a Total NO and Nitrate/Nitrite Parameter Assay Kit (R&D Systems) according to the manufacturer’s instructions.

### 2.8. Statistics Analysis

Statistical analyses were performed using GraphPad Prism v7 (GraphPad Software, La Jolla, CA, USA). Data were presented as mean ± SEM or as the median with interquartile range (IQR) according to the normality of the variables. The Kolmogrov–Smirnov normality test was routinely used. Comparisons of parametric variables between two groups were performed with *T*-test for normally distributed data and Mann–Whitney U test for non-normally distributed data. *p* < 0.05 was considered statistically significant. Correlative studies of NET production and soluble factors were performed using Pearson and Spearman correlation coefficients. Correlation matrices were drawn employing an R package called “corrplot” [23]. The receiver operating characteristic (ROC) curve analysis was also performed to determine the most accurate diagnostic method to discriminate between MPEs and BPEs. Two step multiple lineal regression was applied to predict survival. In the first step, we run a bivariate correlation to screen for variables that were correlated with survival. Then, collinearity of these variables was analyzed. In the second step, multiple linear regression in Enter method with the variables was applied.

## 3. Results

### 3.1. Phenotype of Pleural Neutrophils and Concentration of Neutrophil-Associated Molecules in Pleural Fluids

Absolute numbers of neutrophils were similar in MPE-LAC and PE-HF (Table 2). Flow cytometry analyses revealed that the expression of CD66b, CD16, CD15, and the percentage of CD200R were not statistically different regarding neutrophils from MPE-LAC and PE-HF (Table 2). We also found that the percentages of S100A8/9 positive neutrophils tended to be lower in MPE-LAC than in PE-HF pleural fluids (Table 2). 

IL-8 concentration was significantly higher in MPE-LAC than in PE-HF (Figure 1D). Moreover, MPO, lactoferrin, and MMP-9 levels, representative molecules of primary, secondary, and tertiary granules, respectively, were significantly higher in MPE-LAC than in PE-HF (Figure 1A–C). Additionally, we assessed the possible influence of different age of LAC and HF patients on neutrophil activity. We did not observe any correlation between these neutrophil related molecules and the age of the patients (data not shown). We found that nitrate levels were lower in MPE-LAC than in PE-HF (Figure 1E). Free-DNA was significantly higher in MPE-LAC than in PE-HF (Figure 1F). To assess the diagnostic value of these soluble factors, we performed ROC curves using PE-HF as a reference. The area under the curve (AUC) for MPO, lactoferrin, MMP-9, and IL-8 was 0.781 (95% CI: 0.668-0.87; *p* < 0.001), 0.792 (95% CI: 0.676–0.881; *p* < 0.001), 0.79 (95% CI: 0.678–0.877; *p* < 0.001), and 0.903 (95% CI: 0.81–0.961; *p* < 0.001), respectively. The observed sensitivities were 55.3%, 88.69%, 85.11%, and 93.62% and the specificities were 92%, 62.5%, 72%, and 70.83%, respectively (Appendix A).

### 3.2. Effect of Cell-Free Pleural Fluid on Neutrophil Viability and Function

To assess the influence of pleural fluids on neutrophils, we set up an ex vivo 24h-culture with a conditioned medium supplemented with cf-MPE-LAC and cf-PE-HF. We found that the percentage of viable neutrophils was higher in the presence of cf-MPE-LAC than cf-PE-HF (Figure 2A,B). Regarding neutrophil functions, we showed that neutrophils stimulated in the presence of cf-MPE-LAC released more NETs than in the presence of cf-PE-HF (Figure 2C). We also found that neutrophils stimulated in the presence of cf-MPE-LAC exhibited a reduced oxidative burst compared to cf-PE-HF (Figure 2D,E).

### 3.3. Association between Immunomodulatory Factors in Pleural Fluid and Neutrophil Function

To decipher neutrophil response in MPEs, we correlated neutrophil functions induced by conditioned culture and soluble molecules in pleural fluids. We found that NETosis was positively associated with MMP-9 in both types of pleural fluids (Figure 3A–D). In PE-HF conditioned cultures, NETosis was also positively associated with MPO (Figure 3A,C). In MPE-LAC conditioned cultures, the viability of neutrophils was positively associated with levels of MPO and negatively associated with nitrite (Figure 3B,D). 

We have previously found an altered concentration of platelet derived factors in MPE-LAC. Considering the role of platelets in the modulation of neutrophil response, we analyzed the relationship between PF4, P-selectin, sCD40L, sPD-L1, TGF-β, and VEGF and neutrophil functions. In regard to the NETs induced by conditioned culture, we found a positive correlation with PF4 concentration in cf-PE-HF (Figure 3A,C) and a positive correlation with P-selectin and sPD-L1 in cf-MPE-LAC (Figure 3B,D). We did not find significant associations between oxidative burst and neutrophil viability with platelet-derived factors in either cf-PE-HF patients or cf-MPE-LAC (Figure 3A,B).

### 3.4. Association between Neutrophil Functions and the Clinical Outcome of LAC Patients

We analyzed the relationship between neutrophil functions and inflammation parameters. Pleural levels of LDH and CRP were correlated with the pleural concentration of MPO, MMP-9, and lactoferrin in LAC patients with MPE (Figure 4). However, we did not find any correlation between either LDH or CRP and NETosis or oxidative burst induced by cf-MPE-LAC (data not shown). The presence of tumor cells was positively correlated with pleural levels of MMP-9 (Figure 5A) but negatively correlated with pleural levels of lactoferrin (Figure 5B).

In regard to the clinical outcome, levels of lactoferrin were positively correlated with days of survival after MPE diagnosis (Figure 5C). In contrast, PMA-induced NETosis in conditioned cf-MPE-LAC culture was negatively correlated with days of survival after MPE diagnosis (Figure 5D). We did not find any correlation between days of survival after MPE diagnosis and other degranulation products or oxidative burst (data not shown). A two-step multiple linear regression was run to determine how much variance of survival was explained by lactoferrin and PMA-induced NETosis and their intercorrelated variables (MPO and MMP-9, respectively). These 4 variables significantly predicted survival F(4, 12) = 6.039, *p* < 0.007, adjusted R2 = 0.557). Regarding 180-day survival, a useful time-point in MPE to assess the evolution of these patients [24,25], we found that nonsurvivor patients had a greater capacity to induce NETosis (Figure 5E).

## 4. Discussion

This study shows that MPE-LAC, but not PE-HF, enhance neutrophil survival and NET formation but reduce the oxidative burst capacity of neutrophils. We found higher levels of neutrophil degranulation products in MPE-LAC than in PE-HF. The increased NET production induced by MPEs was associated with higher levels of certain neutrophil- and platelet-derived products and a worse outcome for LAC patients.

We found higher levels of MPO, lactoferrin, and MMP-9 in MPE-LAC than in PE-HF. The ROC curves of these molecules displayed high sensitivity and specificity to discriminate MPE-LAC and PE-HF. Since these molecules are mostly neutrophil derived, it is likely that there is an increased activity of neutrophils in the metastatic pleural compartment of LAC patients. In line with our findings, several reports have shown that MPO and MMP-9 concentrations in neoplastic effusions are higher than in transudates [26,27]. Hegnhoj et al. also found that lactoferrin levels tended to be higher in MPEs from carcinomas and mesotheliomas than in PE-HF [28]. We additionally observed correlations between the concentrations of different degranulation products depending on pleural effusion etiology. Concretely, MMP-9 correlated with MPO and lactoferrin in PE-HF but not in MPE-LAC. It is likely that the release of these three molecules is coordinated in HF but not in LAC. This lack of coordination in MPE may contribute to the dysregulation of NETosis induced by LAC pleural fluids. Additional correlations found between degranulation products and LDH and CRP in MPE-LAC point to the crucial role played by neutrophils in malignancy-induced local inflammation.

Although there are comparable counts of neutrophils in MPE-LAC and PE-HF [5], three current findings, detailed below, would seem to predict higher counts in MPE-LAC than in PE-HF. First, higher levels of degranulation products in MPE-LAC than in PE-HF were seen. However, more neutrophil-derived molecules do not necessarily mean higher counts of neutrophils, but rather more activity per neutrophil. Second, the ROC curve with IL-8 showed the highest sensitivity and specificity to discriminate between MPE-LAC and PE-HF. Nevertheless, like other authors, we found no correlation between the absolute numbers of neutrophils and IL-8 levels (data not shown, [29]). In fact, pleural neutrophil recruitment is an action with complex requirements. Thirdly and finally, neutrophils cultured with MPE-LAC displayed reduced apoptosis. Lifespan may also have affected neutrophil counts. Several molecules in MPEs may be delaying neutrophil apoptosis [30], as indicated by the positive correlation found between levels of MPO and neutrophil viability in MPE-LAC. MPO, mainly produced by neutrophils, can exert a positive feedback via CD11b/CD18 to maintain their survival as previously suggested by El Kebir et al. [31]. Platelets and their derived factors have also been proposed as apoptosis-delaying agents [32], though in the current study, we did not observe a correlation between the viability of neutrophils and platelet-derived mediators. We cannot discard that additional factors, such as GM-CSF, G-CSF, IL-1β, and IFN-γ [33], modulate neutrophil apoptosis in MPE.

Our results indicate that neutrophils perform a different function in the presence of malignant and nonmalignant pleural fluids. It should be highlighted that the different age between HF and LAC patients does not have influence on neutrophil function because experiments were performed with cells from healthy donors. In our approach, we found that NET formation was higher when neutrophils were cultured with cf-MPE-LAC than with cf-PE-HF. This result is consistent with the observed higher levels of extracellular DNA in MPEs than in PE-HF since neutrophils undergoing NETosis are a possible source of extracellular DNA. However, we did not find any correlation between extracellular DNA and NETosis. DNA can also be originated by the death of inflammatory and tumor cells [34], reducing the correlation between DNA extracellular and NETosis. It is interesting to note that Twaddell and colleagues observed more extracellular DNA in malignant pleural fluids than in transudates but without differences in the levels of citrullinated histone 3 [35]. A possible explanation for this discrepancy with our studies is the method used for the quantification of NETs. While we directly quantified NET formation, they measured the levels of a NET protein. Another possible explanation is that we analyzed the functional capacity of PMA-stimulated neutrophils to release NETs in the presence of MPE-LAC while they determined a single marker as a surrogate of NET formation in MPEs.

Positive correlations between MMP-9, P-selectin, and PD-L1 with NETosis in MPE-LAC were found in our study. These findings are not surprising since MMP-9 is a component of DNA webs [36,37]. P-selectin primes neutrophils to undergo NETosis by binding to PSGL-1 [38] and NETs can express PD-L1 [39]. Since we also found a positive correlation between MMP-9 and the percentage of tumor cells, it is tempting to speculate that MMP-9 created a positive feedback, activating neutrophils to release NETs, and thereby favoring tumor progression [40,41]. On the other hand, lactoferrin tended to be negatively correlated with NETosis and, according to Okubo et al., this is possible because lactoferrin can inhibit NET formation without affecting ROS production [42].

We showed that cf-MPE-LAC induced more NETosis but less ROS than cf-PE-HF. This unexpected result has several explanations. It has been reported that one type of NETosis, called lytic NETosis, depends on ROS production by NADPH oxidase complex, though another, called vital NETosis, is faster and NADPH independent [43]. However, in our system we cannot distinguish the NETosis type. It has been described that PMA induces lytic NETosis, but we also cultured neutrophils with cf-MPE-LAC and cf-PE-HF without PMA stimulation and obtained comparable results (data not shown). In addition, the flow cytometry technique that we applied to measure ROS production reflects H_2_O_2_ accumulated inside the cell as a result of the oxidation of DHR to rhodamine [44]. Due to increased levels of MPO, H_2_O_2_ may be more rapidly converted to hypoclorous acid in MPE-LAC than in PE-HF. Other nonexclusive explanations are that NET production and ROS require different incubation times and the contribution of platelet- or tumor-derived exosomes to NET formation [45,46].

We are aware that our study has some limitations. First, we could not directly analyze neutrophils from pleural fluids because they cannot be frozen and thawed. Therefore, our findings with conditioned cultures need further validation with neutrophil pleural fluids. Second, with our experimental system, we cannot determine whether pleural cytokines were originated in pleura microenvironment or if they were transported from circulation. A parallel quantification of plasma and pleural levels in the same patient would provide clues regarding cytokine origin. Third, we cannot ensure about the effect of different anticancer treatments on neutrophils functions, in particular on NETosis as it is frequently unstable. Finally, studies are needed to determine whether our results can be extrapolated to pleural metastases other than lung cancer, such as breast or ovarian carcinomas.

## 5. Conclusions

In conclusion, our study contributes to a better understanding of the mechanisms implicated in the modulation of the neutrophil response in MPEs. In addition, this is the first study associating NETs with a worse outcome in MPE. Our results suggest that NETs themselves may be potential therapeutic targets. It is necessary to explore further the molecular and cellular mechanisms of NETosis in order to develop accurate targeted therapies that block NETosis or their effects on inflammation and facilitate the identification of patients with a worse prognosis.

## Figures and Tables

**Figure 1 cancers-14-02529-f001:**
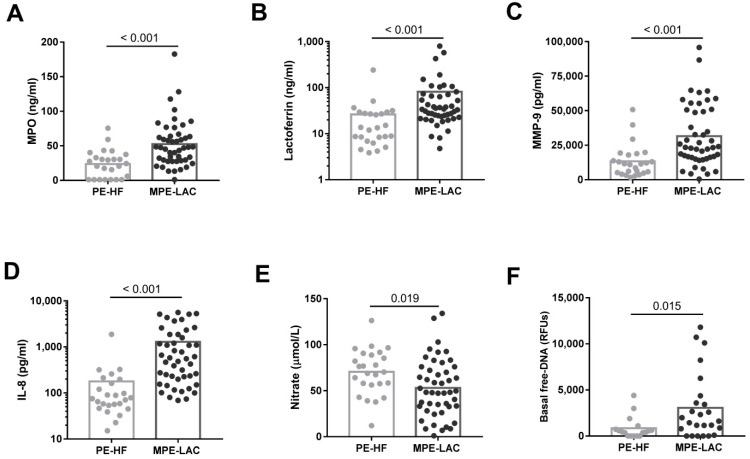
Pleural fluid concentrations of soluble neutrophil-related molecules. Levels of (**A**) myeloperoxidase (MPO; ng/mL), (**B**) lactoferrin (ng/mL), (**C**) matrix metalloproteinase-9 (MMP-9; pg/mL), (**D**) IL-8 (pg/mL), and (**F**) nitrate (µmol/L) in pleural effusions from heart failure (PE-HF) (*n* = 25) and lung adenocarcinoma (MPE-LAC) (*n* = 47) cell-free pleural fluids determined by the corresponding ELISAs. (**F**) Pleural extracellular DNA was measured in PE-HF (*n* = 17) and MPE-LAC (*n* = 25) by fluorescence intensity and expressed as relative fluorescence units (RFUs). The unpaired *T*-test (**E**) and the Mann–Whitney test (**A**–**D**), (**F**) were used for statistical analyses. *P*-values are shown in the graphs.

**Figure 2 cancers-14-02529-f002:**
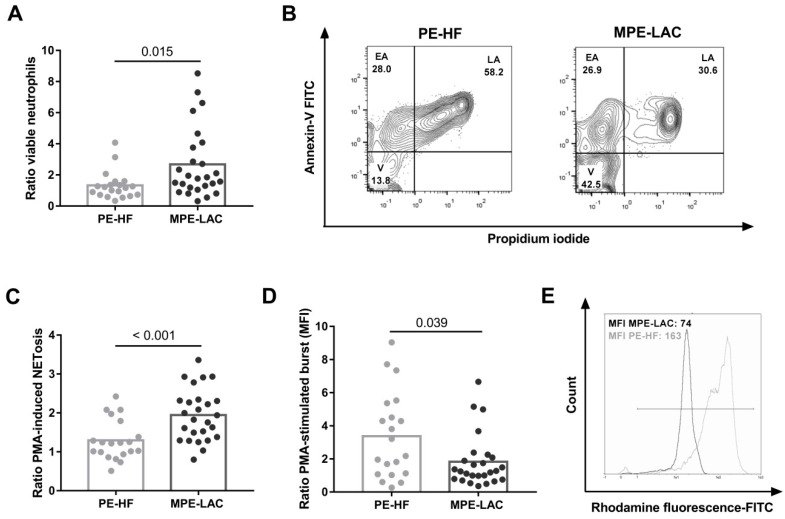
Effect of cell-free pleural fluid on neutrophil viability, NETosis, and oxidative burst. Healthy neutrophils were cultured in the presence of pleural fluids (25% of volume). Viability and oxidative burst capacity were determined by flow cytometry and NET formation was measured by fluorescence. (**A**) The viability of neutrophils in the presence of pleural effusions from heart failure (PE-HF) (*n* = 20) and lung adenocarcinoma (MPE-LAC) (*n* = 26). Data were expressed as ratios calculated by dividing PE-HF or MPE-LAC values by their respective medium values. (**B**) Representative flow cytometry plots of apoptosis assay of neutrophils cultured with cf-PE-HF and cf-MPE-LAC. Quadrants are V for viable cells, EA for early apoptosis, and LA for late apoptotic cells. (**C**,**D**) Capacity of PE-HF (*n* = 20) and MPE-LAC (*n* = 26) to trigger NET formation and oxidative burst, respectively, under PMA-stimulation conditions. Data were expressed as ratios calculated by dividing PE-HF or MPE-LAC values by their respective control medium values. (**E**) Representative flow cytometry histograms of ROS production of neutrophils cultured in the presence of cf-PE-HF (grey line) or MPE-LAC (black line). The Mann–Whitney test was used for statistical analyses. *P*-values are shown in the graphs.

**Figure 3 cancers-14-02529-f003:**
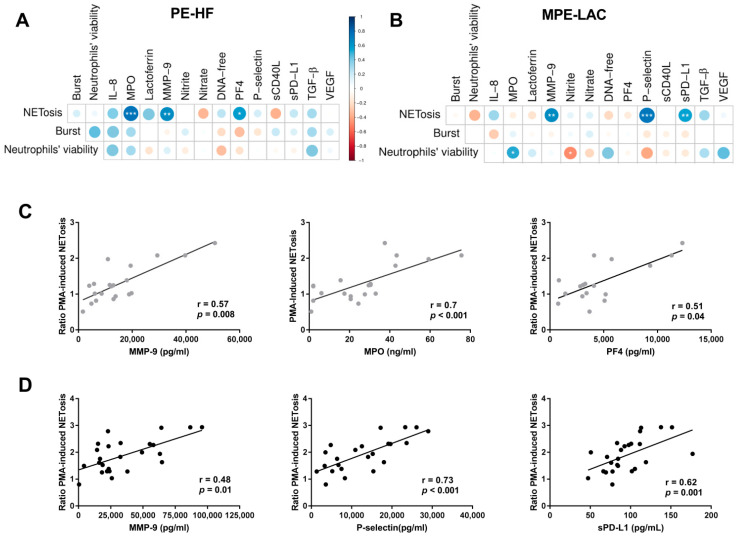
Correlation matrixes between neutrophil functions and platelet and neutrophil-derived factors in cf-PFs. Correlation matrices showing the association between neutrophil- and platelet- derived factors and neutrophils functions in (**A**) PE-HF and (**B**) MPE-LAC. The strength and direction of the correlations are indicated by the circle size and color. The statistical significance was represented by (*) *p* < 0.05, (**) *p* < 0.01, and (***) *p* < 0.001. Spearman correlations were established (**C**) between NETosis and the levels of MPO, MMP-9, and PF4 in cf-PE-HF and (**D**) between NETosis and the levels of lactoferrin and sPD-L1 in cf-MPE-LAC. The Pearson correlation was used for the correlation between NETosis and P-selectin.

**Figure 4 cancers-14-02529-f004:**
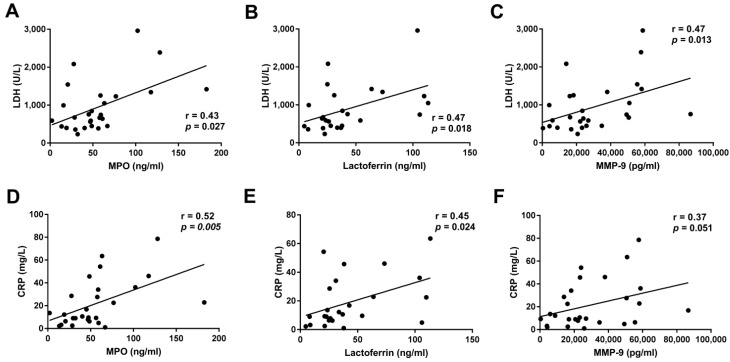
Correlation between paraclinical parameters and the pleural concentration of neutrophil-derived products in LAC patients with MPE. Positive correlation between (**A–C**) LDH (U/L) or (**D–F**) CRP (mg/L) and levels of MPO (ng/mL), lactoferrin (ng/mL) and MMP-9 (pg/mL). *R* and *p* values were calculated using Spearman’s correlation coefficient.

**Figure 5 cancers-14-02529-f005:**
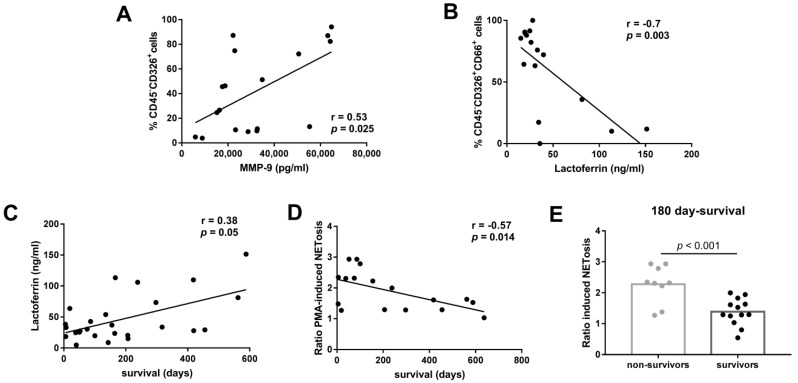
Association between pleural neutrophil-related factors and functions and LAC patient survival. Correlations between the percentage of tumor cells and pleural levels of (**A**) MMP-9 (pg/mL) and (**B**) lactoferrin (ng/mL). (**C**) Spearman positive correlation between pleural lactoferrin levels (ng/mL) and survival days after MPE diagnosis in LAC patients. (**D**) Pearson negative correlation between PMA-induced cf-PE NETosis and survival days after MPE diagnosis. (**E**) Comparison of 180-day survival of LAC patients based on the ability of their pleural fluids to induce NETosis. Correlation coefficients and *p*-values are presented in the graphs. The unpaired *t*-test was used for the statistical analysis of 180-day survival. NETosis-related data are expressed as ratios calculated by dividing LAC values by their respective medium values.

**Table 1 cancers-14-02529-t001:** Demographic, clinical, and molecular parameters of the patients with pleural effusions included in the study.

Patients Characteristics	HF	LAC	*p* Value
Number of patients	25	47	
Male/female	17/8	36/11	n.s.
Median age (range), years	85 (81–88)	69 (62–75)	<0.001
Median (IQR) survival time from diagnosis, days	NA	203.5 (69.3–527.8)	
EGFR mutations	NA	11/47	
ALK mutations	NA	4/47	
**ECOG PS**			
0	NA	12/47	
1	NA	8/47	
2	NA	5/47	
3	NA	10/47	
4	NA	5/47	
NA	NA	7/47	
**Biochemical parameters**			
Glucose (mg/mL)	125.5 (102.8–154)	99 (81.56–123.8)	0.026
Lactate dehydrogenase (U/L)	163.5 (138.8–211.3)	755 (449–1253)	<0.001
C-reactive protein (mg/L)	3.8 (2.3–10.9)	11.2 (6.4–34.6)	0.004
Adenosine deaminase (U/L)	4.3 (3.2–7.8)	7.4 (5.1–10.5)	n.s.
Total protein (g/dL)	2.6 (1.8–3)	4.8 (4–5.7)	<0.001
pH	7.5 ± 0.02	7.4 ± 0.02	0.001
**Treatment**			
Radiotherapy	NA	5/47	
Chemotherapy	NA	17/47	
Immunotherapy	NA	10/47	
ALK inhibitors	NA	2/47	
EGFR inhibitors	NA	5/47	
Her2 inhibitors	NA	2/47	
Untreated or palliative care	NA	6/47	
**Soluble platelet- related molecules in pleural fluids**			
PF4 (pg/mL)	3281 (2125–4593)	8282 (4107–14,605)	<0.001
VEGF (ng/mL)	119.3 (1–455.3)	2377 (313.3–3598)	0.001
P-selectin (pg/mL)	4440 ± 323.7	11,484 ± 1260	0.001
TGF-β (pg/mL)	11,572 (9320–15,950)	16,838 (8662–28,772)	0.04
CD40L (pg/mL)	2.1 (1–56.2)	26.2 (2.1–157.1)	0.07
PD-L1 (pg/mL)	40.3 (32.5–73.6)	96.4 (76.4–132.6)	<0.001

ALK: anaplastic lymphoma kinase; EGFR: epidermal growth factor receptor; HF, heart failure; LAC, lung adenocarcinoma; NA, not applicable; n.s., non-significant; PF4, platelet factor 4, CD40L, soluble CD40L; PD-L1, soluble programmed death-ligand 1; TGF-β, transforming growth factor-β; VEGF, vascular endothelial growth factor. Nonparametric Mann–Whitney test was used for all comparisons except for pH and P-selectin, which were compared by *t*-test.

**Table 2 cancers-14-02529-t002:** Phenotype of neutrophils present in pleural effusions from HF and LAC.

	PE-HF	MPE-LAC	*p* Value
Neutrophils (cell/mL)	24,000 (4967–106,120)	21,170 (1491–140,580)	n.s.^1^
Neutrophil phenotype			
CD15 expression (MFI)	48.6 ± 6.5	55.9 ± 10.9	n.s. ^2^
CD16 expression (MFI)	21.8 ± 3.8	26.2 ± 3.6	n.s. ^2^
CD66b expression (MFI)	19.2 (14.2–25.3)	22.9 (14.7–26.8)	n.s. ^1^
S100A8/9^+^ neutrophils (%)	48.5 ± 7.4	31.4 ± 3.8	0.09 ^2^
CD200R^+^ neutrophils (%)	5.1 ± 1.9	4.3 ± 0.6	n.s. ^2^

MFI: mean fluorescence intensity. ^1^ Nonparametric Mann–Whitney test; ^2^
*t*-test.

## Data Availability

All data relevant to the study are included in the article or uploaded as Appendix A. The datasets used and analyzed during the current study are included in the article. They are available from the corresponding author on reasonable request.

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
