# Peer review of "Influence of Malignant Pleural Fluid from Lung Adenocarcinoma Patients on Neutrophil Response"

_cancers, 2022, doi:10.3390/cancers14102529_

Round 1
Reviewer 1 Report
Neutrophils are the most abundant circulating leukocytes in humans. They develop from myeloid precursors and play a central role in the innate immune response by destroying foreign particles either intracellularly in phagosomes or extracellularly by releasing neutrophil extracellular traps (NETs), and promoting acute inflammation. In the tumor microenvironment (TME), these immune cells were described as pro-tumorigenic or anti-tumorigenic, depending on the inflammatory mediators present in the TME. The pro-tumorigenic function has been related to different pathways including the formation of NETs that can induce NETosis (the cell death induced by NET) and protect tumor cells from cytotoxic lymphocytes. The anti-tumor action of neutrophils was related to the production of oxidative components such as ROS and NO. Neutrophils were described as the most prevalent immune cell type in NSCLC. In advanced cancer, malignant pleural effusion (MPE) was produced and a high proportion of neutrophils in MPE was associated with shorter survival in lung cancer patients. So neutrophils have a potential role as therapeutic targets. Deciphering the mechanisms of the pro-tumor/pro-metastatic role of neutrophils and neutrophil plasticity will facilitate developing neutrophils-targeting agents and novel therapeutic strategies for patients with advanced disease in the near future.
In order to characterize the neutrophils in pleural fluid, the authors compared the population of neutrophils in begnin pleural fluid obtained from patient with heart failure (HF) and in malignant pleural fluid obtained from patients diagnosed with lung adenocarcinoma (LAC). They measured the proportion of neutrophils in both pleural fluids using flow cytometry. They characterized their activity by measuring the degranulation products (MPO, lactoferrin, MMP9) secreted by neutrophils and the NETosis induced by NET (measuring the free-DNA in pleural fluid). They also studied how the microenvironment of the pleural fluid can affect the activity and function of neutrophils. To this end, they cultured isolated neutrophils from the blood of healthy donors in media supplemented (or not) with pleural fluid from HF or LAC patients. Then they studied the viability and activity of the neutrophils.
They showed that pleural fluids from LAC patients contained the same amount of neutrophils as pleural fluid from HF patients. But malignant pleural fluid contained higher levels of the pro-inflammatory cytokine IL-8, and the degranulation products (MPO, lactoferrin, MMP9). The degranulation product levels were related to inflammatory markers including LDH and CRP. MPE from LAC patients contained a higher level of free-DNA compared to HC patients, suggesting a higher NETosis activity of the neutrophils. Then the authors demonstrated that components of the pleural fluid from LAC patients increased the viability of neutrophils, and their capacity to form NET but reduced their oxidative burst activity. The correlation study of the components of the pleural fluid and the neutrophil functions showed different associations in pleural fluid from HF and LAC patients. In contrast to HF patients, the viability of neutrophils in LAC patients was positively associated with MPO and negatively associated with Nitrite. Their NETosis activity was positively associated with the levels of MMP-9, P-selectin and sPD-L1. In HF patients, the NETosis activity of neutrophils was also positively associated with the levels of MMP-9, but not P-selectin and sPD-L1.
The study was well constructed and the methods used were suitable. The article is clearly written, and easy to read.
I have some question/remarks:
- the median age and range of both cohorts of patients were significantly different. This is a major problem because you cannot exclude that the differences observed between these two groups are not due to age. Discuss this point in the discussion part.
How do you explain that the pleural extracellular DNA has increased in LAC pleural fluids and that the % of S100A8/9+ neutrophils in LAC was decreased (compared to HF)?
Minor points:
MATERIAL AND METHODS
Line 125-128 – the isolation of neutrophils from healthy donors was not described even in the reference cited. This point should be improved.
Lines 133 and 147: the PMA concentration used should be stated in the same unit (nM or ng/ml).
Line 287 – Figure 5 showed correlations between the percentage of tumor cells and pleural levels of different components. Indicate in the “Material and methods part”, how the tumor cells were identified and quantified.
RESULTS
Table 1: the Range of “Median (IQR) survival time from diagnosis, days is wrong.
Figures
Line 281: Add in the legend of figure 4 that the parameters studied were from LAC patients.
DISCUSSION
Line 308: The author’s name of the reference 26 is not Graudal. Correct it.
Author Response
We really appreciate the effort of reviewers to assess the first version of manuscript 1713599: “Association of NETosis in malignant pleural fluids and poor outcomes in lung adenocarcinoma patients”. It has been reviewed according to their comments and suggestions. Here, we are submitting a document in response to the reviewer's comments and the second version of the document with the corresponding changed figures and new information.

Reviewer 2 Report
The current paper is interesting in it's attempt to clarify basic mechanisms in the innate immune response in pleural effusion, with focus on malignant pleural effusion secondary to metastatic lung adenocarcinoma.
I'm a clinician and clinical researcher, so I'm not able to dissect the in vivo aspects. My focus is on the clinical setting.
MAJOR CONCERNS
1) Focus
The paper is framed (introduction line 49) as an attempt to identify new biomarkers in MPE. That is not obvious that this is what this study does. Rather this study unravels the role of neutrophils in MPE secondary to lung adenocarcinoma. The sentence on challenges to separate MPE from BPE (lines 46-50) appears irrelevant in this paper's Introduction, as this topic is in no way the scope of the present study. Or if it is, please respond to it in Results stating that "Neutrophil products cannot be used to differentiate MPE from HF". Or delete lines 46-50 to avoid to confuse the reader.
The authors focus is clearly on cellular mechanisms. Thus the description of included patients is very sparse, and limited to a few lines in Table 1, stating that the heart failure (HF) group was markedly older than the malignant pleural effusion group (MPE). As the title is "Association of NETosis in malignant pleural fluids and poor outcomes in lung adenocarcinoma patients", the reader deserves information on performance status which is a well-documented clinical biomarker for poor outcome. So is elevated levels of systemic CRP or leukocytes. You may consider to change your title to the main scope: comparison of NETosis between HF and MPE.
2) Primary endpoint
Primary endpoint according to Title is "Median (IQR) survival time
from diagnosis, days = 203.5 (699.3‐527.8)" (Table 1). Firstly, how can IQR go from a higher to a lower value and why was this presented as IQR and not minimum-maximum? Secondly, NETosis is hardly stable from MPE diagnosis to death, so please add information on how many days from MPE diagnosis that the pleural effusion was collected for this study. Please consider if NETosis level was a consequence of time living with MPE rather than a predictor for death. Thirdly, in Figure 5 c-d: how were these parameters chosen among the very many variables analysed? Please state if these associations were the only significant and thus hand-picked among the many negative. It's fair to do se but then it should be stated and a Bonferroni correction should be added.
3) Patient groups
Correct definition of the two clinical groups is pivotal to draw any conclusions:
"Heart Failure" is not defined
Unclear if "LAC" is identical to MPE = cytological verification of malignant cells in pleural fluid derived from a pulmonary adenocarcinoma? LAC is not a common abbrevation whereas MPE is widely used. I suggest to replace LAC with MPE or MPE-lac for readers' clarity.
4) Obvious confounders
The authors do not discuss that NETosis is age dependent (see https://www.ncbi.nlm.nih.gov/pmc/articles/PMC5820386/), with a weakened response in the very elderly (aka the HF group). As the Devil's advocate, it is easy to dismiss all inter-group differences in neutrophil count and activity by the marked age difference. Please convince me and future readers that this is not the case.
Likewise the Limitations section (lines 375-383) is not exhaustive. Please add: external validity to other cancers, effect on NETosis on various anti-cancer drugs and other drugs, stability of NETosis in the individual patient, and does NETosis measurement add anything as a biomarker (which is the framing of NETosis in Introduction, line 49) for death compared to the clinical evaluation of performance status.
MINOR COMMENTS
Methods: page 2
Table 1: inconsistent use of decimals throughout. Age in days with decimals: the decimal is clinically irrelevant and does not add any information.
Table 1, biochemistry: unclear if parentheses ressemble "range" or "interquartile range". HF group PF4: no range reported. HF group P-selectin +/-standard deviation ? MPE P-selectin: no range or SD reported.
Table 1 statistics: CD40L appears from range not to be normally distributed. Recommend to use non-parametric tests throughout.
Results,
All Tables and Figures: heterogeneous use of HF, LAC, HF patients, and LAC patients. Suggest HF and MPE, and not "HF patients", as the politically correct term is "patients with HF". Please change.
Table 2: mean+/- SEM involves many negative values. Is this physiologically possible or does it reflect that data are not normally distributed? Were data normally distributed logarithmically?
Table 1, treatment options are not mutually exclusive. What is relevant is if patients receive ongoing treatment, which may affect neutrophil activity. So you could reduce to "ongoing systemic treatment yes" and "ongoing systemic treatment no".
Figure 4: pleural CRP or LDH are not clinical parameters but paraclinical parameters. Please alter text to Figure 4.
Figure 5e: a cut-off of 180 days Survival is not a well-defined clinical outcome, please add a reference for this decision.
Author Response

(The authors gave the same response as above.)

Reviewer 3 Report
Reviewer comments:
Comments to the Author
The authors have investigated in Malignant pleural effusion (MPE), NETosis was positively associated with MMP‐9, P‐selectin, and sPD‐L1 and clinically related to a worse outcome. The findings suggest an association of NETs with a worse outcome in MPE that may lead to tumor progression through the release of NETs, suggesting that they are a potential therapeutic target in lung adenocarcinoma.
- Mulet et al., present an interesting manuscript where association of NETs with the worsen outcome in MPE has been described. The study is well planned and logically presented.
- All in all, I found the experimental procedures used very helpful in demonstrating this important physio-patholoigical link between: NETs -> outcomes in MPE.
Round 2
Reviewer 2 Report
Congratulations with a satisfying revision of your impressive research paper.
I have only one suggestion: consider to change the title further so it covers both malignant (MPE) and benign (HF) pleural effusion. In the Discussion you write: "Our results indicate that neutrophils perform a different function in the presence of malignant and non-malignant pleural fluids", so a new title could be "Different neutrophil response in malignant and non-malignant pleural effusion".